# Neural Solvers for Fast and Accurate Numerical Optimal Control

**Federico Berto**
KAIST, `DiffEqML`
`fberto@kaist.ac.kr`

**Stefano Massaroli**
The University of Tokyo, `DiffEqML`
`massaroli@robot.t.u-tokyo.ac.jp`

**Michael Poli**
Stanford University, `DiffEqML`
`zymrael@cs.stanford.edu`

**Jinkyoo Park**
KAIST
`jinkyoo.park@kaist.ac.kr`

## ABSTRACT

Synthesizing optimal controllers for dynamical systems often involves solving optimization problems with hard real–time constraints. These constraints determine the class of numerical methods that can be applied: computationally expensive but accurate numerical routines are replaced by fast and inaccurate methods, trading inference time for solution accuracy. This paper provides techniques to improve the quality of optimized control policies given a fixed computational budget. We achieve the above via a *hypersolvers* (Poli et al., 2020a) approach, which hybridizes a differential equation solver and a neural network. The performance is evaluated in *direct* and *receding–horizon* optimal control tasks in both low and high dimensions, where the proposed approach shows consistent Pareto improvements in solution accuracy and control performance.

## 1 INTRODUCTION

Optimal control of complex, high–dimensional systems requires computationally expensive numerical methods for *differential equations* (Pytlak, 2006; Rao, 2009). Here, real–time and hardware constraints preclude the use of accurate and expensive methods, forcing instead the application of cheaper and less accurate algorithms. While the paradigm of optimal control has successfully been applied in various domains (Vadali et al., 1999; Lewis et al., 2012; Zhang et al., 2016), improving accuracy while satisfying computational budget constraints is still a great challenge (Ross & Fahroo, 2006; Baotić et al., 2008). To alleviate computational overheads, we detail a procedure for *offline* optimization and subsequent *online* application of hypersolvers (Poli et al., 2020a) to optimal control problems. These hybrid solvers achieve the accuracy of higher–order methods by augmenting numerical results of a base solver with a learning component trained to approximate local truncation residuals. When the cost of a single forward–pass of the learning component is kept sufficiently small, hypersolvers improve the computation–accuracy Pareto front of low–order explicit solvers (Butcher, 1997). However, direct application of hybrid solvers to controlled dynamical system involves learning truncation residuals on the higher–dimensional spaces of state and control inputs. To extend the range of applicability of hypersolvers to controlled dynamical systems, we propose two pretraining strategies designed to improve, in the set of admissible control inputs, on the average or worst–case hypersolver solution. With the proposed methodology, we empirically show that Pareto front improvements of hypersolvers hold even for optimal control tasks. In particular, we then carry out performance and generalization evaluations in direct and model predictive control tasks. Here, we confirm Pareto front improvements in terms of solution accuracy and subsequent control performance, leading to higher quality control policies and lower control losses. In high–dimensional regimes, we obtain the same control policy as the one obtained by accurate high–order solvers with more than $3\times$ speedup.

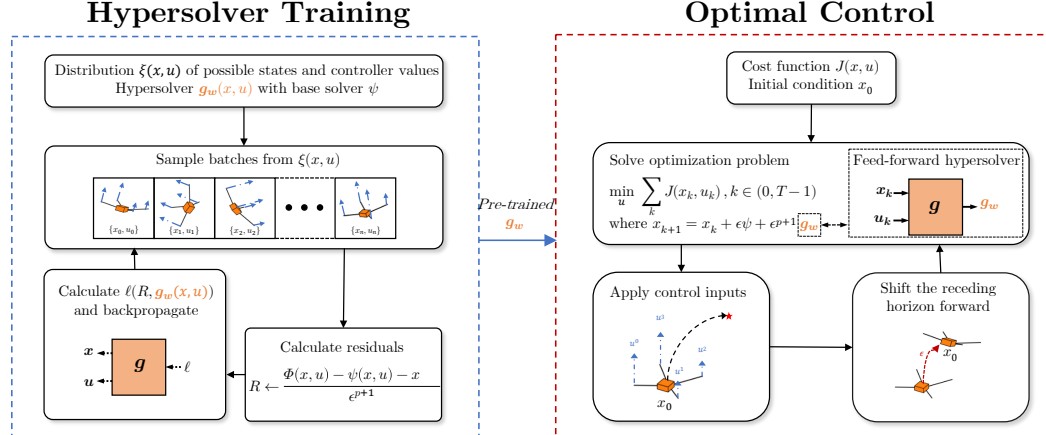

Figure 1: Overview of the proposed method. [Left] The hypersolver is trained to approximate residuals given a distribution of control inputs and states. [Right] The pre–trained hypersolver model is then used to accelerate and improve the accuracy of numerical solutions used during optimization of control policies, leading to higher–quality controllers.

## 2 NUMERICAL OPTIMAL CONTROL

We consider control of general nonlinear systems of the form

$$
\begin{aligned}
\dot{x}(t) &= f(t, x(t), u_\theta(t)) \\
x(0) &= x_0
\end{aligned}
\tag{1}
$$

with state $x \in \mathcal{X} \subset \mathbb{R}^{n_x}$, input $u_\theta \in \mathcal{U} \subset \mathbb{R}^{n_u}$ defined on a compact time domain $\mathcal{T} := [t_0, T]$ where $\theta$ is a finite set of free parameters of the controller. Solutions of (1) are denoted with $x(t) = \Phi(x(s), s, t)$ for all $s, t \in \mathcal{T}$. Given some objective function $J : \mathcal{X} \times \mathcal{U} \to \mathbb{R}$; $x_0, u_\theta \mapsto J(x_0, u_\theta(t))$ and a distribution $\rho_0(x_0)$ of initial conditions with support in $\mathcal{X}$, we consider the following nonlinear program, constrained to the system dynamics:

$$
\begin{aligned}
\min_{u_\theta(t)} \quad & \mathbb{E}_{x_0 \sim \rho_0(x_0)} \left[ J(x_0, u_\theta(t)) \right] \\
\text{subject to} \quad & \dot{x}(t) = f(t, x(t), u_\theta(t)) \\
& x(0) = x_0 \\
& t \in \mathcal{T}
\end{aligned}
\tag{2}
$$

where the controller parameters $\theta$ are optimized. We will henceforth omit the subscript $\theta$ and write $u(t) = u_\theta(t)$. Since analytic solutions of (2) exist only for limited classes of systems and objectives, numerical solvers are often applied to iteratively find a solution. For these reasons, problem 2 is often referred to as *numerical optimal control*.

**Direct optimal control**  If the problem (2) is solved *offline* by directly optimizing over complete trajectories, we call it *direct* optimal control. The infinite–dimensional optimal control problem is time–discretized and solved numerically: the obtained control policy is then applied to the real target system without further optimization.

**Model predictive control**  Also known in the literature as receding horizon control, Model Predictive Control (MPC) is a class of flexible control algorithms capable of taking into consideration constraints and nonlinearities (Mayne & Michalska, 1988; Garcia et al., 1989). MPC considers finite time windows which are then shifted forward in a *receding* manner. The control problem is then solved for each window by iteratively forward–propagating trajectories with numerical solvers i.e. *predicting* the set of future trajectories with a candidate controller $u(t)$ and then adjusting it iteratively to optimize the cost function $J$ (further details on the MPC formulation in Appendix B.2). The optimization is reiterated *online* until the end of the control time horizon.

## 2.1 SOLVER RESIDUALS

Given nominal solutions $\Phi$ of (1) we can define the *residual* of a numerical ODE solver as the normalized error accumulated in a single step size of the method, i.e.

$$R_k = R(t_k, x(t_k), u(t_k)) = \frac{1}{\epsilon^{p+1}}\Big[\Phi(x(t_k), t_k, t_{k+1}) - x(t_k) - \epsilon\psi_\epsilon(t_k, x(t_k), u(t_k))\Big] \qquad (3)$$

where $\epsilon$ is the step size and $p$ is the order of the numerical solver corresponding to $\psi_\epsilon$. From the definition of residual in (3), we can define the *local truncation error* $e_k := \big\|\epsilon^{p+1}R_k\big\|_2$ which is the error accumulated in a single step; while the *global truncation error* $\mathcal{E}_k = \|x(t_k) - x_k\|_2$ represents the error accumulated in the first $k$ steps of the numerical solution. Given a $p$–th order explicit solver, we have $e_k = \mathcal{O}(\epsilon^{p+1})$ and $\mathcal{E}_k = \mathcal{O}(\epsilon^p)$ (Butcher, 1997).

## 3 HYPERSOLVERS FOR OPTIMAL CONTROL

We extend the range of applicability of hypersolvers (Poli et al., 2020a) to controlled dynamical systems. In this Section we discuss the proposed hypersolver architectures and pre–training strategies of the proposed hypersolver methodology for numerical optimal control of controlled dynamical systems.

## 3.1 HYPERSOLVERS

Given a $p$–order base solver update map $\psi_\epsilon$, the corresponding *hypersolver* is the discrete iteration

$$x_{k+1} = x_k + \underbrace{\epsilon\psi_\epsilon\left(t_k, x_k, u_k\right)}_{\text{base solver step}} + \epsilon^{p+1}\underbrace{g_\omega\left(t_k, x_k, u_k\right)}_{\text{approximator}} \qquad (4)$$

where $g_\omega\left(t_k, x_k, u_k\right)$ is some $o(1)$ parametric function with free parameters $\omega$. The core idea is to select $g_\omega$ as some function with *universal approximation* properties and fit the higher-order terms of the base solver by explicitly minimizing the residuals over a set of state and control input samples. This procedure leads to a reduction of the overall *local truncation error* $e_k$, i.e. we can improve the base solver accuracy with the only computational overhead of evaluating the function $g_\omega$. It is also proven that, if $g_\omega$ is a $\delta$–approximator of $R$, i.e. $\forall k \in \mathbb{N}_{\leq K}$

$$\|R\left(t_k, x(t_k), u(t_k)\right) - g_\omega\left(t_k, x(t_k), u(t_k)\right)\|_2 \leq \delta \qquad (5)$$

then $e_k \leq o(\delta\epsilon^{p+1})$, where $\delta > 0$ depends on the hypersolver training results (Poli et al., 2020a, Theorem 1). This result practically guarantees that if $g_\omega$ is a good approximator for $R$, i.e. $\delta \ll 1$, then the overall local truncation error of the *hypersolved* ODE is significantly reduced with guaranteed upper bounds.

## 3.2 NUMERICAL OPTIMAL CONTROL WITH HYPERSOLVERS

Our approach relies on the pre–trained hypersolver model for obtaining solutions to the trajectories of the optimal control problem (2). After the initial training stage, control policies are numerically optimized to minimize the cost function $J$ (see Appendix B.3 for further details). Figure 1 shows an overview of the proposed approach consisting in pre–training and system control.

## 4 HYPERSOLVER PRE–TRAINING AND ARCHITECTURES

We introduce in Section 4.1 loss functions which are used in the proposed pre–training methods of Section 4.2 and Section 4.3. We also check the generalization properties of hypersolvers with different architectures in Section 4.4. In Section 4.5 we introduce *multi–stage hypersolvers* in which an additional first–order learned term is employed for correcting errors in the vector field.

## 4.1 LOSS FUNCTIONS

**Residual fitting** Training the hypersolver on a single nominal trajectory $\{x(t_k)\}_k$ results in a *supervised* learning problem where we minimize point–wise the Euclidean distance between the

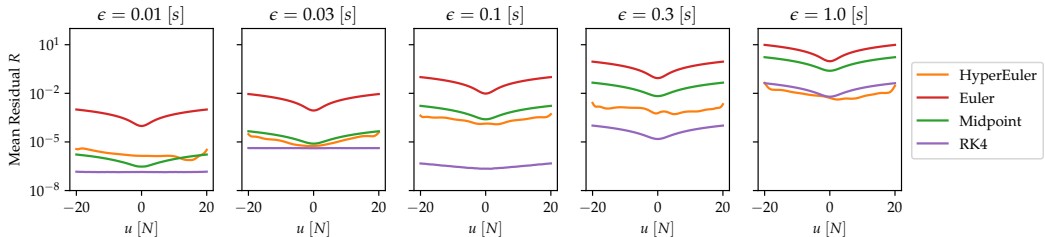

Figure 2: Mean local residuals of the spring–mass system of (17) as a function of control inputs at different step sizes $\epsilon$. HyperEuler (see Appendix A.1 for its explicit formulation) improves on the local residuals compared to the baseline Euler and even compared to higher-order ODE solvers at larger step sizes.

residual (3) and the output of $g_\omega$, resulting in an optimization problem minimizing a loss function $\ell$ of the form

$$\ell(t, x, u) = \frac{1}{K} \sum_{k=0}^{K-1} \|R(t_k, x(t_k), u(t_k)) - g_\omega(t_k, x(t_k), u(t_k))\|_2 \tag{6}$$

which is also called *residual fitting* since the target of $g_w$ is the residual $R$.

**Trajectory fitting**  The optimization can also be carried out via *trajectory fitting* as following

$$\ell(t, x, u) = \frac{1}{K} \sum_{k=0}^{K-1} \|x(t_{k+1}), x_{k+1}\|_2 \tag{7}$$

where $x(t_{k+1})$ corresponds to the exact one–step trajectory and $x_{k+1}$ is its approximation, derived via (4) for standard hypersolvers or via (11) for their multi–stage counterparts. This method can also be used to contain the global truncation error in the $\mathcal{T}$ domain. We will refer to $\ell$ as a loss function of either residual or trajectory fitting types; we note that these loss functions may also be combined depending on the application. The goal is to train the hypersolver network to explore the state–control spaces so that it can effectively minimize the truncation error. We propose two methods with different purposes: *stochastic exploration* aiming at minimizing the average truncation error and *active error minimization* whose goal is to reduce the maximum error i.e., due to control inputs yielding high losses.

## 4.2 Stochastic Exploration

Stochastic exploration aims to minimize the *average* error of the visited state–controller space i.e., to produce optimal hypersolver parameters $\omega^*$ as the solution of a nonlinear program

$$\omega^* = \arg\min_\omega \mathbb{E}_{\xi(x,u)}[\ell(t, x, u)] \tag{8}$$

where $\xi(x, u)$ is a distribution with support in $\mathcal{X} \times \mathcal{U}$ of the state and controller spaces and $\ell$ is the training loss function. In order to guarantee sufficient exploration of the state–controller space, we use Monte Carlo sampling (Robert & Casella, 2013) from the given distribution. In particular, batches of initial conditions $\{x_0^i\}$, $\{u_0^i\}$ are sampled from $\xi$ and the loss function $\ell$ is calculated with the given system and step size $\epsilon$. We then perform backpropagation for updating the parameters of the hypersolver using a *stochastic gradient descent* (SGD) algorithm e.g., Adam (Kingma & Ba, 2017) and repeat the procedure for every training epoch. Figure 2 shows pre–training results with stochastic exploration for different step sizes (see Appendix C.2). We notice how higher residual values generally correspond to higher absolute values of control inputs. Many systems in practice are subject to controls that are constrained in magnitude either due to physical limitations of the actuators or safety restraints of the workspace. This property allows us to design an exploration strategy that focuses on worst-case scenarios i.e. largest control inputs.

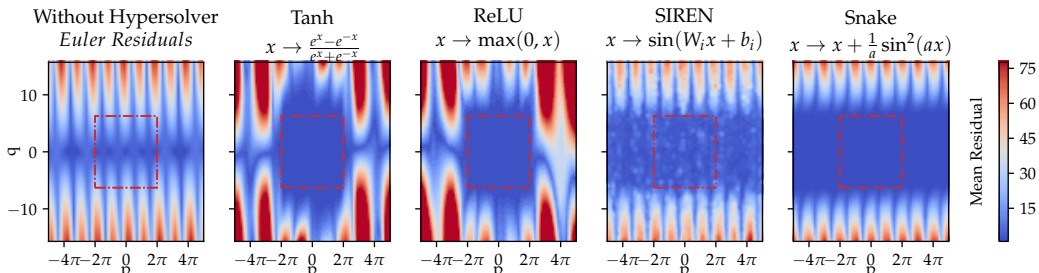

Figure 4: Generalization outside of the training region (red rectangle) in the state space of an inverted pendulum model with different hypersolver activation functions. Architectures containing activation functions with periodic components achieve better extrapolation properties compared to the others.

## 4.3 ACTIVE ERROR MINIMIZATION

The goal of active error minimization is to actively reduce the highest losses in terms of the control inputs, i.e., to obtain $w^*$ as the solution to a minmax problem:

$$w^* = \arg\min_{\omega} \ \max_{u \in \mathcal{U}} \ \ell(t, x, u) \qquad (9)$$

Similarly to stochastic exploration, we create distribution $\xi(x, u)$ with support in $\mathcal{X} \times \mathcal{U}$ and perform Monte Carlo sampling of $n$ batches $\{(x^i, u^i)\}$, from $\xi$. Then, losses are computed pair–wisely for each state $x^j$, $j = 0, \ldots, n-1$ with each control input $u^k$, $k = 0, \ldots, n-1$. We then take the first $n$ controllers $\{u_0^{i'}\}$ yielding the maximum loss for each state. The loss is recalculated using these controller values with their respective states and SGD updates to hypersolver parameters are performed. Figure 3 shows a comparison of the pre–training techniques (further experimental details in Appendix C.2). The propagated error on trajectories for the hypersolver pre–trained via stochastic exploration is lower on average with random control inputs compared with the one pre–trained with active error minimization. The latter accumulates lower error for controllers yielding high residuals.

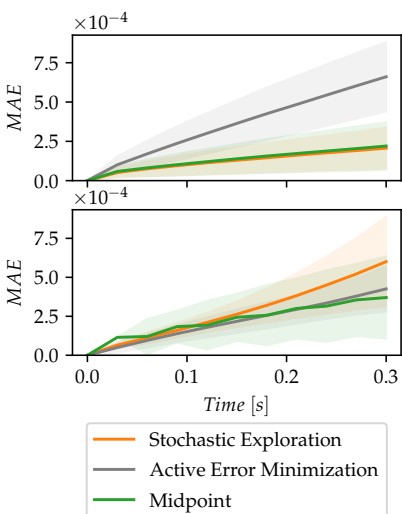

Figure 3: *Mean Absolute Error* (MAE) along trajectories with different pre–training techniques on the spring–mass system of (17). [Top] Stochastic exploration performs better on average i.e. $u \in [-100, 100]$. [Bottom] Active error minimization achieves better results in limit situations as in the case of a *bang–bang controller* i.e. $u \in \{-100, 100\}$, in which controllers yielding the highest residuals have been minimized.

> Different exploration strategies may be used depending on the down–stream control task.

## 4.4 GENERALIZATION PROPERTIES OF DIFFERENT ARCHITECTURES

We have assumed the state and controller spaces to be bounded and that training be performed by sampling for their known distributions. While this is sufficient for optimal control problems given *a priori* known bounds, we also investigate how the system generalizes to unseen states and control input values. In particular, we found that activation functions have an impact on the end result of generalization beyond training boundaries. We take into consideration two commonly used activation functions, Tanh : $x \to \frac{e^x - e^{-x}}{e^x + e^{-x}}$ and ReLU : $x \to \max(0, x)$, along with network architectures which employ activation functions containing periodic components: SIREN : $x \to \sin(Wx + b)$ (Sitzmann et al., 2020) and Snake : $x \to x + \frac{1}{a}\sin^2(ax)$ (Ziyin et al., 2020). We train hypersolver models with the different activation functions for the inverted pendulum model of (18) with common experimental settings (see Appendix C.3). Figure 4 shows generalization

outside the training states (see Figure 9 in Appendix C.3 for generalization of controllers and step sizes). We notice that while `Tanh` and `ReLU` perform well on the training set of interest, performance degrades rapidly outside of it. On the other one hand, `SIREN` and `Snake` manage to extrapolate the periodicity of the residual distribution even outside of the training region, thus providing further empirical evidence of the universal extrapolation theorem (Ziyin et al., 2020, Theorem 3).

> Activation function choice plays an important role in Hypersolver performance and generalization.

## 4.5 Multi–stage Hypersolvers

We have so far considered the case in which the vector field (1) fully characterizes the system dynamics. However, if the model does not completely describe the actual system dynamics, first–order errors are introduced. We propose `Multi-Stage Hypersolvers` to correct these errors: an additional term is introduced in order to correct the inaccurate dynamics $f$. The resulting procedure is a modified version of (4) in which the base solver $\psi_\epsilon(t_k, x_k, u_k)$ does not iterate over the modeled vector field $f$ but over its corrected version $f^\star$:

$$f^\star(t_k, x_k, u_k) = \underbrace{f(t_k, x_k, u_k)}_{\text{partial dynamics}} + \underbrace{h_w(t_k, x_k, u_k)}_{\text{inner stage}} \tag{10}$$

where $h_w$ is a function with universal approximation properties. While the *inner stage* $h_w$ is a first–order error approximator, the *outer stage* $g_\omega$ further reduces errors approximating the $p$–th order residuals:

$$x_{k+1} = x_k + \epsilon \psi_\epsilon \left( t_k, x_k, u_k, \underbrace{f^\star(t_k, x_k, u_k)}_{\text{corrected dynamics}} \right) + \epsilon^{p+1} \underbrace{g_\omega(t_k, x_k, u_k)}_{\text{outer stage}} \tag{11}$$

We note that $f^\star$ is continuously adjusted due to the optimization of $h_w$. For this reason, it is not possible to derive the analytical expression of the residuals to train the stages with the residual fitting loss function (6). Instead, both stages can be optimized at the same time via backpropagation calculated on one–step trajectory fitting loss (7) which does not require explicit residuals calculation.

## 5 Experiments

We introduce the experimental results divided for each system into hypersolver pre–training and subsequent optimal control. We use as accurate adaptive step–size solvers the Dormand/Prince method `dopri5` (Dormand & Prince, 1980) and an improved version of it by Tsitouras `tsit5` (Tsitouras, 2011) for training the hypersolvers and to test the control performance at runtime.

### 5.1 Direct optimal control of a Pendulum

**Hypersolver pre–training** We consider the inverted pendulum model with a torsional spring described in (18). We select $\xi(x, u)$ as a uniform distribution with support in $\mathcal{X} \times \mathcal{U}$ where $\mathcal{X} = [-2\pi, 2\pi] \times [-2\pi, 2\pi]$ and $\mathcal{U} = [-5, 5]$ to guarantee sufficient exploration of the state-controller space. Nominal solutions are calculated using `tsit5` with absolute and relative tolerances set to $10^{-5}$. We train the hypersolver on local residuals via stochastic exploration using the `Adam` optimizer with learning rate of $3 \times 10^{-4}$ for $3 \times 10^5$ epochs.

**Direct optimal control** The goal is to stabilize the inverted pendulum in the vertical position $x^\star = [0, 0]$. We choose $t \in [0, 3]$ and a step size $\epsilon = 0.2\ s$ for the experiment. The control input is assumed continuously time–varying. The neural controller is optimized via SGD with `Adam` with learning rate of $3 \times 10^{-3}$ for 1000 epochs. Figure 5 shows nominal controlled trajectories of HyperEuler and other baseline fixed–step size solvers. Trajectories obtained with the controller optimized with HyperEuler reach final positions $q = (1.6 \pm 17.6) \times 10^{-2}$ while Midpoint and RK4 ones $q = (-0.6 \pm 12.7) \times 10^{-2}$ and $q = (1.1 \pm 12.8) \times 10^{-2}$ respectively. On the other hand, the controller optimized with the Euler solver fails to control some trajectories obtaining a final

$q = (6.6 \pm 19.4) \times 10^{-1}$. HyperEuler considerably improved on the Euler baseline while requiring only $1.2\%$ more *Floating Point Operations* (FLOPs) and $49.5\%$ less compared to Midpoint. Further details are available in Appendix C.3.

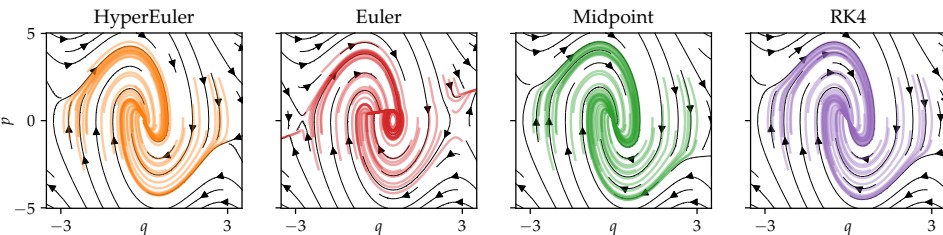

Figure 5: Direct optimal control of the inverted pendulum in phase space. While the controller optimized with the Euler solver fails to control the system for some trajectories, the one obtained with HyperEuler improves the performance while introducing a minimal overhead with results comparable to higher–order solvers.

## 5.2 MODEL PREDICTIVE CONTROL OF A CART-POLE SYSTEM

**Hypersolver pre–training** We consider the partial dynamics of the cart–pole system of (19) with wrong parameters for the frictions between cart and track as well as the one between cart and pole. We employ the multi–stage hypersolver approach to correct the first–order error in the vector field as well as base solver residual. We select $\xi(x, u)$ as a uniform distribution with support in $\mathcal{X} \times \mathcal{U}$ where $\mathcal{X} = [-2\pi, 2\pi] \times [-2\pi, 2\pi] \times [-2\pi, 2\pi] \times [-2\pi, 2\pi]$ and $\mathcal{U} = [-10, 10]$. Nominal solutions are calculated on the accurate system using `RungeKutta 4` instead of adaptive–step solvers due faster training times. We train our multi–stage Hypersolver (i.e. a multi–stage hypersolver with the second–order Midpoint as base solver with the partial dynamics) on nominal trajectories of the accurate system via stochastic exploration using the `Adam` optimizer for $5 \times 10^4$ epochs, where we set the learning rate to $10^{-2}$ for the first $3 \times 10^4$ epochs, then decrease it to $10^{-3}$ for $10^4$ epochs and to $10^{-4}$ for the last $10^4$.

**Model predictive control** The goal is to stabilize the cart–pole system in the vertical position around the origin, e.g. $x^\star = [0, 0, 0, 0]$. We choose $t \in [0, 3]$ and a step size $\epsilon = 0.05 \, s$ for the experiment. The control input is assumed piece-wise constant during MPC sampling times. The receding horizon is chosen as $1 \, s$. The neural controller is optimized via SGD with `Adam` with learning rate of $3 \times 10^{-3}$ for a maximum of 200 iterations at each sampling time. Figure 6 shows nominal controlled trajectories of multi–stage Hypersolver and other baseline solvers. The Midpoint solver on the inaccurate model fails to stabilize the system at the origin position $x = (39.7 \pm 97.7) \, cm$, while multi–stage Hypersolver manages to stabilize the cart–pole system and improve on final positions $x = (7.8 \pm 3.0) \, cm$. Further details are available in Appendix C.4.

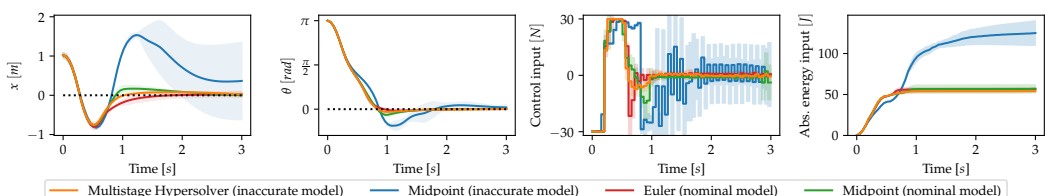

Figure 6: Model Predictive Control with constrained inputs on the cart–pole model. MPC with the Midpoint solver iterating on the partial dynamic model successfully swings up the pole but fails to reach the target position. Multi–stage Hypersolver with the Midpoint base solver has knowledge restricted to the inaccurate system, yet it manages to obtain a similar control performance compared to controllers with access to the nominal dynamics while also needing less control effort and absolute energy inflow compared to its base solver.

> Multi–stage Hypersolvers can correct first–order errors on dynamic models and base solver residuals.

### 5.3 Model Predictive Control of a Quadcopter

**Hypersolver pre–training** We consider the quadcopter model of (20). We select $\xi(x, u)$ as a uniform distribution with support in $\mathcal{X} \times \mathcal{U}$ where $\mathcal{X}$ is chosen as a distribution of possible visited states and each of the four motors $i \in [0, 3]$ has control inputs $u^i \in [0, 2.17] \times 10^5$ rpm. Nominal solutions are calculated on the accurate system using dopri5 with relative and absolute tolerances set to $10^{-7}$ and $10^{-9}$ respectively. We train HyperEuler on local residuals via stochastic exploration using the Adam optimizer with learning rate of $10^{-3}$ for $10^5$ epochs.

**Model predictive control** The control goal is to reach a final positions $[x, y, z]^\star = [8, 8, 8]\ m$. We choose $t \in [0, 3]$ and a step size $\epsilon = 0.02\ s$ for the experiment. The control input is assumed piece–wise constant during MPC sampling times. The receding horizon is chosen as $0.5\ s$. The neural controller is optimized via SGD with Adam with learning rate of $10^{-2}$ for 20 iterations at each sampling time. Figure 7 shows local residual distribution and control performance on the quadcopter over 30 experiments starting at random initial conditions which are kept common for the different ODE solvers. HyperEuler requires a single function evaluation per step as for the Euler solver compared to two function evaluations per step for Midpoint and four for RK4. Controlled trajectories optimized with Euler, Midpoint and RK4 collect an error on final positions of $(1.09 \pm 0.37)\ m$, $(0.71 \pm 0.17)\ m$, $(0.70 \pm 0.19)\ m$ respectively while HyperEuler achieves the lowest terminal error value of $(0.66 \pm 0.24)\ m$. Additional experimental details are available in Appendix C.5.

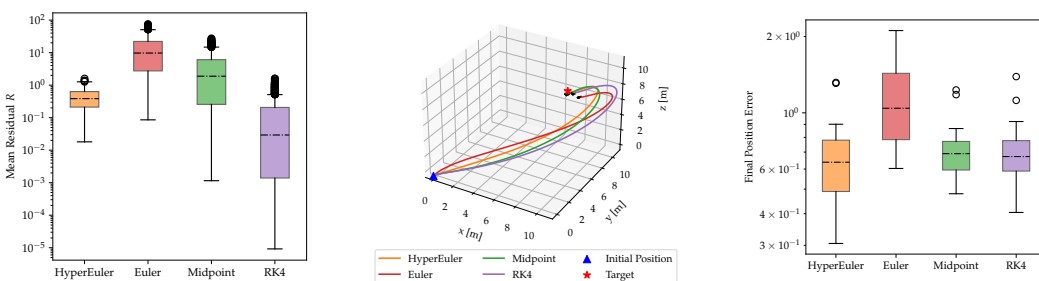

Figure 7: [Left] Local residual distribution for the quadcopter model for $\epsilon = 0.02\ s$. [Center] Trajectories of controlled quadcopters with MPC whose receding horizon controller is optimized by solving the ODE with different methods. [Right] Final positions error distribution. The proposed approach with HyperEuler achieves lower average error compared to other baseline solvers while requiring a low overhead compared to higher–order solvers due to a smaller number of dynamics function evaluations.

### 5.4 Boundary Control of a Timoshenko Beam

**Hypersolver pre–training** We consider the finite element discretization of the Timoshenko beam of (22). We create $\xi(x, u)$ as a distribution with support in $\mathcal{X} \times \mathcal{U}$ which is generated at training time via random walks from known boundary conditions in order to guarantee both physical feasibility and sufficient exploration of the state-controller space (see Appendix C.6 for further details). Nominal solutions are calculated using tsit5 with absolute and relative tolerances set to $10^{-5}$. We train the hypersolver on local residuals via stochastic exploration using the Adam optimizer for $10^5$ epochs, where we set the learning rate to $10^{-3}$ for the first $8 \times 10^4$ epochs, then decrease it to $10^{-4}$ for $10^4$ epochs and to $10^{-5}$ for the last $10^4$.

**Boundary direct optimal control** The task is to stabilize the beam in the straight position, i.e. each of its elements $i$ have velocities $v_t^i, v_r^i$ and displacements $\sigma_t^i, \sigma_r^i$ equal to 0. We choose $t \in [0, 3]$ and step size $\epsilon = 5\ ms$ for the experiment. The control input is assumed continuously time–varying. The neural controller is optimized via SGD with Adam with learning rate of $10^{-3}$ for 1000 epochs. Figure 8 shows nominal controlled trajectories for HyperEuler and other baseline fixed–step size solvers. Control policies trained with Euler and Midpoint obtain averaged final states of $(-2.8 \pm 4.2) \times 10^{-1}$ and $(-0.04 \pm 4.6) \times 10^{-1}$ thus failing to stabilize the beam, while HyperEuler and RK4 obtain $(-0.6 \pm 4.9) \times 10^{-3}$ and $(-0.5 \pm 3.3) \times 10^{-3}$ respectively. HyperEuler considerably improves on both the Euler and Midpoint baselines obtaining a very similar performance to RK4, while requiring $72.9\%$ less FLOPs; the mean runtime per training iteration was cut from $8.24\ s$ for RK4 to just $2.53\ s$ for HyperEuler. Further details on this experiment are available in Appendix C.6.

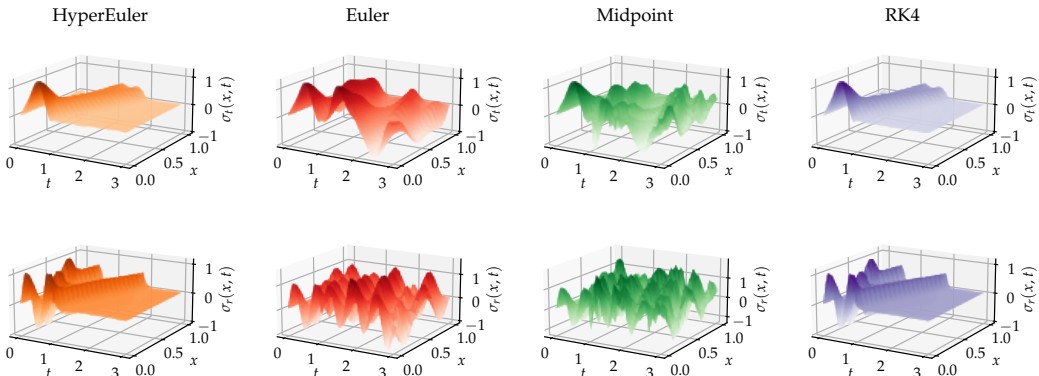

Figure 8: Displacement variables $\sigma_t$ and $\sigma_r$ of the discretized Timoshenko beam as a function of position $x$ of the finite elements and time $t$. The controller optimized with HyperEuler manages to stabilize the beam while the baseline solvers Euler and Midpoint fail, yet requiring less than a third in terms of runtime compared to RK4.

> Hypersolvers are even more impactful in complex high–dimensional controlled systems.

## 6   RELATED WORK

This work is rooted in the broader literature on surrogate methods for speeding up simulations and solutions of dynamical systems (Grzeszczuk et al., 1998; James & Fatahalian, 2003; Gorissen et al., 2010). Differently from these approaches, we investigate a methodology to enable faster solution during a downstream, online optimization problem involving a potential mismatch compared to data seen during pre–training. We achieve this through the application of the *hypersolver* (Poli et al., 2020a) paradigm. Modeling mismatches between approximate and nominal models is explored in (Saveriano et al., 2017) where residual dynamics are learned efficiently along with the control policy while (Fisac et al., 2018; Taylor et al., 2019) model systems uncertainties in the context of safety–critical control. In contrast to previous work, we model uncertainties with the proposed multi–stage hypersolver approach by closely interacting with the underlying ODE base solvers and their residuals to improve solution accuracy. The synergy between machine learning and optimal control continues a long line of research on introducing neural networks in optimal control (Hunt et al., 1992), applied to modeling (Lin & Cunningham, 1995), identification (Chu et al., 1990) or parametrization of the controller itself (Lin et al., 1991). Existing surrogate methods for systems (Grzeszczuk et al., 1998; James & Fatahalian, 2003) pay a computational cost upfront to accelerate downstream simulation. However, ensuring transfer from offline optimization to the online setting is still an open problem. In our approach, we investigate several strategies for an accurate offline–online transfer of a given hypersolver, depending on desiderata on its performance in terms of average residuals and error propagation on the online application. Beyond hypersolvers, our approach further leverages the latest advances in hardware and machine learning software (Paszke et al., 2019) by solving thousands of ODEs in parallel on *graphics processing units (GPUs)*.

## 7   CONCLUSION

We presented a novel method for obtaining fast and accurate control policies. Hypersolver models were firstly pre–trained on distributions of states and controllers to approximate higher–order residuals of base fixed–step ODE solvers. The obtained models were then employed to improve the accuracy of trajectory solutions over which control policies were optimized. We verified that our method shows consistent improvements in the accuracy of ODE solutions and thus on the quality of control policies optimized through numerical solutions of the system. We envision the proposed approach to benefit the control field and robotics in both simulated and potentially real–world environments by efficiently solving high–dimensional space–continuous problems.

## CODE OF ETHICS

We acknowledge that all the authors of this work have read and commit to adhering to the ICLR Code of Ethics.

## REPRODUCIBILITY STATEMENT

We share the code used in this paper and make it publicly available on Github[1]. The following appendix also supplements the main text by providing additional clarifications. In particular, Appendix A provides further details on the considered hypersolver models. We provide additional information on optimal control policy in Appendix B while in Appendix C we provide details on on the system dynamics, architectures and other experimental details. Additional explanations are also provided as comments in the shared code implementation.

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

## A  ADDITIONAL HYPERSOLVER MATERIAL

### A.1  EXPLICIT HYPEREULER FORMULATION

Our analysis in the experiments takes into consideration the *hypersolved* version of the Euler scheme, namely HyperEuler. Since Euler is a first–order method, it requires the least number of function evaluations (NFE) of the vector field $f$ in (1) and yields a second order local truncation error $e_k := \left\| \epsilon^2 R_k \right\|_2$. This error is larger than other fixed–step solvers and thus has the most room for potential improvements. The base solver scheme $\psi_\epsilon$ of (4) can be written as $\psi_\epsilon (t_k, x_k, u_k) = f (t_k, x_k, u_k)$, which is approximating the next state by adding an evaluation of the vector field multiplied by the step size $\epsilon$. We can write the HyperEuler update explicitly as

$$x_{k+1} = x_k + \epsilon f(t_k, x_k, u_k) + \epsilon^2 g_w (t_k, x_k, u_k) \tag{12}$$

while we write its residual as

$$R (x(t_k), u(t_k))) = \frac{1}{\epsilon^2} (\Phi(x(t_k), t_k, t_{k+1}) - x(t_k) - \epsilon f(t_k, x_k, u_k)) \tag{13}$$

### A.2  HYPERSOLVERS FOR TIME–INVARIANT SYSTEMS

A *time–invariant* system with time–invariant controller can be described as following

$$\dot{x}(t) = f(x(t), u(x(t)))$$
$$x(0) = x_0 \tag{14}$$

in which $f$ and $u$ do not explicitly depend on time. The models considered in the experiments satisfy this property.

## B  CONTROL POLICY DETAILS

### B.1  OPTIMAL CONTROL COST FUNCTION

The general form of the *integral cost functional* can be written as follows

$$J(x(t), u(t)) = [x^\top (t_f) - x^\star] \mathbf{P}[x(t_f) - x^\star] + \int_{t_0}^{t_f} \left( [x^\top (t) x^\star] \mathbf{Q}[x(t) - x^\star] + u^\top (t) \mathbf{R} u(t) \right) dt \tag{15}$$

where matrix $\mathbf{P}$ is a penalty on deviations from the target $x^\star$ of the last states, $\mathbf{Q}$ penalizes all deviations from the target of intermediate states while $\mathbf{R}$ is a regulator for the control inputs. Evaluation of (15) usually requires numerical solvers such as the proposed hypersolvers of this work. Discretizations of the cost functional are also called *cost function* in the literature.

### B.2  MODEL PREDICTIVE CONTROL FORMULATION

The following problem is solved *online* and iteratively until the end of the time span

$$\begin{aligned} \min_{u_k} \quad & \sum_{k=0}^{T-1} J (x_k, u_k) \\ \text{subject to} \quad & \dot{x}(t) = f(t, x(t), u(t)) \\ & x(0) = x_0 \\ & t \in \mathcal{T} \end{aligned} \tag{16}$$

where $J$ is a cost function and $T \in \mathcal{T}$ is the *receding horizon* over which predicted future trajectories are optimized.

### B.3  NEURAL OPTIMAL CONTROL

We parametrize the control policy of problem (2) as $u_\theta : t, x \mapsto u_\theta(t, x)$ where $\theta$ is a finite set of free parameters. Specifically, we consider the case of *neural* optimal control in which controller $u_\theta$ is a multi–layer perceptron. The optimal control task is to minimize the cost function $J$ described in (15) and we do so by optimizing the parameters $\theta$ via SGD; in particular, we use the Adam (Kingma & Ba, 2017) optimizer for all the experiments.

## C  Experimental Details

In this section we include additional modeling and experimental details divided into the different dynamical systems.

### C.1  Hypersolver Network Architecture

We design the hypersolver networks $g_w$ as feed–forward neural networks. Table 1 summarizes the parameters used for the considered controlled systems, where *Activation* denotes the activation functions, i.e. SoftPlus: $x \mapsto \log(1 + e^x)$, Tanh: $x \mapsto \frac{e^x - e^{-x}}{e^x + e^{-x}}$ and Snake : $x \to x + \frac{1}{a}\sin^2(ax)$ (Ziyin et al., 2020). We also use the vector field $f$ as an input of the hypersolver, which does not

Table 1: Hyper–parameters for the hypersolver networks in the experiments.

|  | Spring–Mass | Inverted Pendulum [2] | Cart–Pole[3] | Quadcopter | Timoshenko Beam |
|---|---|---|---|---|---|
| Input Layer | 5 | 5 | 9 | 28 | 322 |
| Hidden Layer 1 | 32 | 32 | 32 | 64 | 256 |
| Activation 1 | Softplus | Softplus | Snake | Softplus | Snake |
| Hidden Layer 2 | 32 | 32 | 32 | 64 | 256 |
| Activation 2 | Tanh | Tanh | Snake | Softplus | Snake |
| Output Layer | 2 | 2 | 4 | 12 | 160 |

require a further evaluation since it is pre–evaluated at runtime by the base solver $\psi$. We emphasize that the size of the network should depend on the application: a too–large neural network may require more computations than just increasing the numerical solver's order: Pareto optimality of hypersolvers also depends on their complexity. Keeping their neural network small enough guarantees that evaluating the hypersolvers is cheaper than resorting to more complex numerical routines.

### C.2  Spring-mass System

**System Dynamics**  The spring-mass system considered is described in the Hamiltonian formulation by

$$\begin{bmatrix} \dot{q} \\ \dot{p} \end{bmatrix} = \begin{bmatrix} 0 & 1/m \\ -k & 0 \end{bmatrix} \begin{bmatrix} q \\ p \end{bmatrix} + \begin{bmatrix} 0 \\ 1 \end{bmatrix} u \tag{17}$$

where $m = 1\ [Kg]$ and $k = 0.5\ [N/m]$.

**Pre-training methods comparison**  We select $\xi(x, u)$ as a uniform distribution with support in $\mathcal{X} \times \mathcal{U}$ where $\mathcal{X} = [-20, 20] \times [-20, 20]$ while $\mathcal{U} = [-100, 100]$. Nominal solutions are calculated on the accurate system using dopri5 with relative and absolute tolerances set to $10^{-7}$ and $10^{-9}$ respectively. We train two separate HyperEuler models with different training methods on local residual for step size $\epsilon = 0.03\ s$: stochastic exploration and active error minimization. The optimizer used is Adam with learning rate of $10^{-3}$ for $10^4$ epochs.

**Hypersolvers with different step sizes**  We select $\xi(x, u)$ as a uniform distribution with support in $\mathcal{X} \times \mathcal{U}$ where $\mathcal{X} = [-5, 5] \times [-5, 5]$ while $\mathcal{U} = [-20, 20]$. Nominal solutions are calculated on the accurate system using dopri5 with relative and absolute tolerances set to $10^{-7}$ and $10^{-9}$ respectively. We train separate HyperEuler models with stochastic exploration with different step sizes $\epsilon$. The optimizer used is Adam with learning rate of $10^{-3}$ for $10^4$ epochs.

---

[2]This architecture refers to the optimal control experiment. Details on hypersolver models for the generalization experiment on the inverted pendulum are available in Appendix C.3.

[3]In the multi–stage hypersolver experiment we consider both the inner stage $h_w$ and the outer stage $g_\omega$ with the same architecture and jointly trained (more information and ablation study in Appendix C.4).

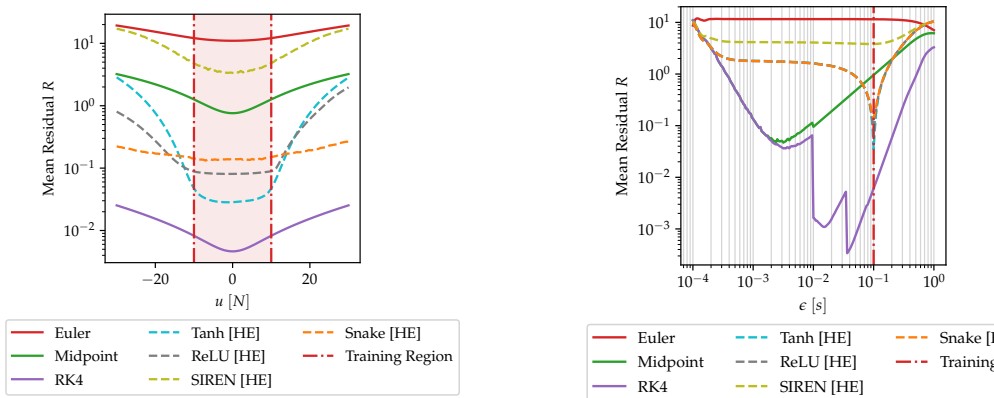

Figure 9: Generalization with different hypersolver activation functions (HyperEuler models are marked with "HE") on the inverted pendulum. [Left] Generalization for the controller space outside of the training region (red area). The architecture with Snake can to generalize better compared to other hypersolvers. [Right] Generalization for different time steps outside of the training step $\epsilon = 0.1\ s$ (red line). HyperEuler is able to improve the baseline Euler solver performance even for unseen $\epsilon$.

## C.3 INVERTED PENDULUM

**System Dynamics**  We model the inverted pendulum with elastic joint with Hamiltonian dynamics via the following:

$$\begin{bmatrix} \dot{q} \\ \dot{p} \end{bmatrix} = \begin{bmatrix} 0 & 1/m \\ -k & -\beta/m \end{bmatrix} \begin{bmatrix} q \\ p \end{bmatrix} - \begin{bmatrix} 0 \\ mgl\sin q \end{bmatrix} + \begin{bmatrix} 0 \\ 1 \end{bmatrix} u \tag{18}$$

where $m = 1\ [Kg]$, $k = 0.5\ [N/\mathrm{rad}]$, $r = 1\ [m]$, $\beta = 0.01\ [Ns/\mathrm{rad}]$, $g = 9.81\ [m/s^2]$.

**Pre–training for the generalization study**  We perform sampling via stochastic exploration from the uniform distribution $\xi(x, u)$ with support in $\mathcal{X} \times \mathcal{U}$ with $\mathcal{X} = [-2\pi, 2\pi] \times [-2\pi, 2\pi]$ and $\mathcal{U} = [-10, 10]$ for the different architectures. We choose as a common time step $\epsilon = 0.1\ s$; the networks are trained for 100000 epochs with the Adam optimizer and learning rate of $10^{-3}$. The network architectures share the same parameters as the inverted pendulum ones in 1, while the activation functions are substituted by the ones in Figure 4. The SIREN architecture is chosen with 2 hidden layers of size 64. Figure 9 provides an additional empirical results on generalization properties across controller values and step sizes: we notice how Snake can generalize to unseen control values better compared to other hypersolvers.

**Additional visualization**  Figure 10 provides an additional visualization of the inverted pendulum controlled trajectories from Figure 5 with positions $q$ and momenta $p$ over time.

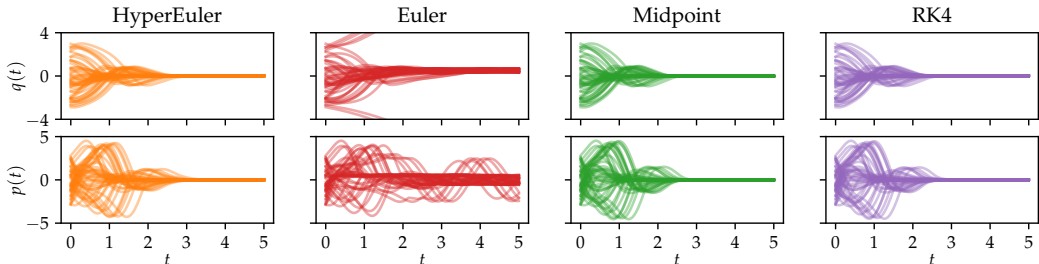

Figure 10: Controlled trajectories of the inverted pendulum with controllers optimized via different solvers.

## C.4 CART-POLE

**System Dynamics**  We consider a continuous version of a cart–pole system additionally taking into account the full dynamic model in Florian (2005). This formulation considers the friction coefficient

between the track and the cart $\mu_c$ inducing a force opposing the linear motion as well as the friction generated between the cart and the pole $\mu_p$, whose generated torque opposes the angular motion. The full cart–pole model is described by the four variables $x, \dot{x}, \theta, \dot{\theta}$ and the accelerations update is as following

$$N_c = (m_c + m_p) g - m_p l \left( \ddot{\theta} \sin \theta + \dot{\theta}^2 \cos \theta \right)$$

$$\ddot{\theta} = \frac{g \sin \theta + \cos \theta \left[ \frac{-u - m_p l \dot{\theta}^2 (\sin \theta + \mu_c \; \mathrm{sgn}(N_c \dot{x}) \cos \theta)}{m_c + m_p} + \mu_c g \; \mathrm{sgn}(N_c \dot{x}) \right] - \frac{\mu_p \dot{\theta}}{m_p l}}{l \left[ \frac{4}{3} - \frac{m_p \cos \theta}{m_c + m_p} \left( \cos \theta - \mu_c \; \mathrm{sgn}(N_c \dot{x}) \right) \right]} \tag{19}$$

$$\ddot{x} = \frac{u + m_p l \left( \dot{\theta}^2 \sin \theta - \ddot{\theta} \cos \theta \right) - \mu_c N_c}{m_c + m_p}$$

where $m_c = 1 \; [Kg]$, $m_p = 0.1 \; [Kg]$, $l = 0.5 \; [m]$ and $g = 9.81 \; [m/s^2]$. $N_c$ represents the *normal force* acting on the cart. For simulation purposes, we consider its sign to be always positive when evaluating the sign (sgn) function as the cart should normally not jump off the track. Setting $\mu_c, \mu_p$ to 0 results in the same dynamic model used in the OpenAI Gym (Brockman et al., 2016) implementation.

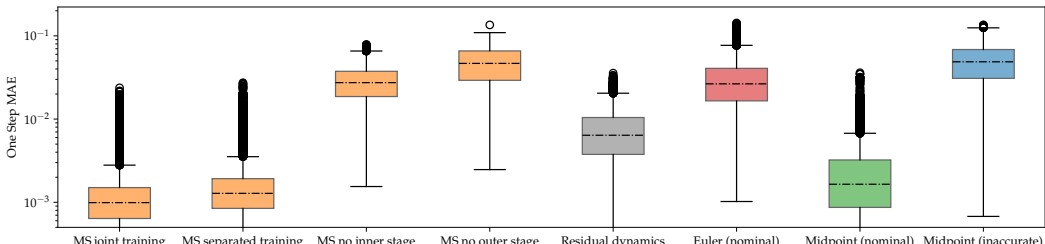

Figure 11: One step *Mean Absolute Error* (MAE) for multi–stage hypersolvers and different solvers as well as correction schemes in the ablation study. Multi–stage Hypersolver (MS) with joint training and Midpoint base solver iterating on an inaccurate vector field agnostic of friction forces outperforms the Midpoint solver with full knowledge of the vector field.

**Multistage training strategies** We study two different training strategies for the inner and outer networks $h_w$ and $g_\omega$ in (11). We first consider a *joint training* strategy in which both stages are trained at the same time via stochastic exploration. Secondly, we do a *separated training* in which only the inner stage network $h_w$ is trained first and then the outer stage network $g_\omega$ is added and only its parameters are trained in a *finetuning* process. We find that, as shown in Figure 11, joint training yields slightly better results. A further advantage of jointly training both stages is that only a single training procedure is required.

**Ablation study** We consider the same model as our Multi–stage Hypersolver with base Midpoint solver but no first–stage $h_w$, which corresponds to learning the *residual dynamics* only, and we train this model with stochastic exploration. We show in Figure 11 that while the residual dynamics model can improve the one–step error compared to the base solver on the inaccurate dynamics, it performs worse the Multi–stage Hypersolver scheme. We additionally study the contribution of each stage in the prediction error improvements by separately zeroing out the contributions of the inner and outer stage. While iterating over the inner stage only improves on the base–solver error, including the outer stage further contributes in improving the error. We notice how the excluding the inner–stage yields higher errors: this may be due to the fact that the inner–stage specializes in correcting the first–order vector field inaccuracies while the outer–stage corrects the one step base solver residual.

**Additional experimental details** For the Multi–stage Hypersolver control experiment, we pre–train both inner and outer stage networks $h_w$ and $g_\omega$ in (11) at the same time using stochastic exploration. The base solver is chosen as the second–order Midpoint iterating on the partial dynamics (19) with $\mu_c, \mu_p$ set to 0. The nominal dynamics considers non–null friction forces: we set the cart friction coefficient to $\mu_c = 0.1$ and the one of the pole to $\mu_p = 0.03$. We note how the friction coefficients make the vector field (19) *non–smooth*: simulation through adaptive–step size

solvers as `tsit5` results experimentally time–consuming, hence we resort to `RK4` for training the hypersolver networks. Nonetheless, as shown in the error propagation of Figure 12, this does not degrade the performance of the trained multi–stage hypersolver scheme. All neural networks in the experiements, including the ablation study, are trained with the `Adam` optimizer for $5 \times 10^4$ epochs, where we set the learning rate to $10^{-2}$ for the first $3 \times 10^4$ epochs, then decrease it to $10^{-3}$ for $10^4$ epochs and to $10^{-4}$ for the last $10^4$.

## C.5 QUADCOPTER

**System Dynamics**   The quadcopter model is a suitably modified version of the explicit dynamics update in (Panerati et al., 2021) for batched training in PyTorch. The following accelerations update describes the dynamic model

$$\ddot{\mathbf{x}} = \left( \mathbf{R} \cdot [0, 0, k_F \textstyle\sum_{i=0}^{3} \omega_i^2] - [0, 0, mg] \right) m^{-1}$$
$$\ddot{\boldsymbol{\psi}} = \mathbf{J}^{-1} \left( \tau(l, k_F, k_T, [\omega_0^2, \omega_1^2, \omega_2^2, \omega_3^2]) - \dot{\boldsymbol{\psi}} \times \left( \mathbf{J}\dot{\boldsymbol{\psi}} \right) \right) \tag{20}$$

mwhere $\mathbf{x} = [x, y, z]$ corresponds to the drone positions and $\boldsymbol{\psi} = [\phi, \theta, \psi]$ to its angular positions; $\boldsymbol{R}$ and $\boldsymbol{J}$ are its rotation and inertial matrices respectively, $\tau(\cdot)$ is a function calculating the torques induced by the motor speeds $\omega_i$, while arm length $l$, mass $m$, gravity acceleration constant $g$ along with $k_F$ and $k_T$ are scalar variables describing the quadcopter's physical properties.

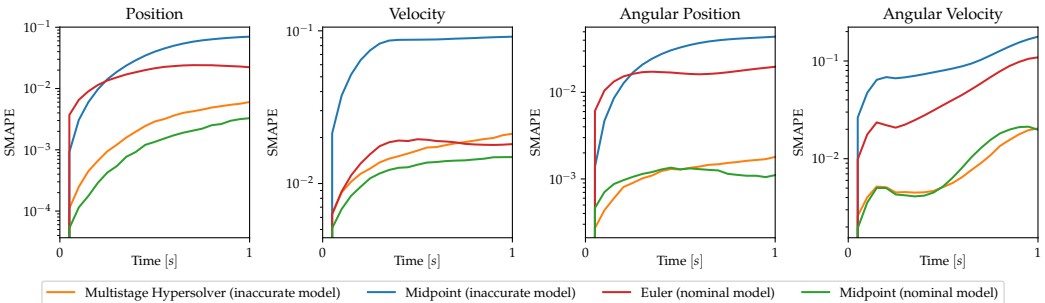

Figure 12: *Symmetric Mean Absolute Percentage Error* (SMAPE) propagation along controlled trajectories of the cart–pole system. The Multi–stage Hypersolver with knowledge limited to the inaccurate model manages to outperform the Euler solver iterating on the accurate dynamics in terms of positions and angular positions.

## C.6 TIMOSHENKO BEAM

**System Dynamics**   We consider as a system from the theory of continuum dynamics the Timoshenko beam with no dissipation described in (Macchelli & Melchiorri, 2004; Massaroli et al., 2021). The system can be described in the coenergy formulation by the following *partial differential equation* (PDE)

$$\begin{bmatrix} \rho A & 0 & 0 & 0 \\ 0 & I_\rho & 0 & 0 \\ 0 & 0 & C_b & 0 \\ 0 & 0 & 0 & C_s \end{bmatrix} \frac{\partial}{\partial t} \begin{pmatrix} v_t \\ v_r \\ \sigma_r \\ \sigma_t \end{pmatrix} = \begin{bmatrix} 0 & 0 & 0 & \partial_x \\ 0 & 0 & \partial_x & 1 \\ 0 & \partial_x & 0 & 0 \\ \partial_x & -1 & 0 & 0 \end{bmatrix} \begin{pmatrix} v_t \\ v_r \\ \sigma_r \\ \sigma_t \end{pmatrix} \tag{21}$$

where $\rho$ is the mass density, $A$ is the cross section area, $I_\rho$ is the rotational inertia, $C_s$ and $C_b$ are the shear and bending compliance; the discretized state variables $v_t$, $v_r$ represent the translational and rotational velocities respectively while $\sigma_t, \sigma_r$ denote the translational and rotational displacements. cantilever beam. In order to discretize the problem, we implement a software routine based on the `fenics` (Alnæs et al., 2015) open–source software suite to obtain the finite–elements discretization of the Timoshenko PDE of (21) given the number of elements, physical parameters of the model and initial conditions of the beam. We choose a 40 elements discretization of the PDE for a total of 160 dimensions of the discretized state $z = [v_t, v_r, \sigma_t, \sigma_r]^\top$ and we initialize the beam at time $t = 0$ as $z(x, 0) = [\sin(\pi x), \sin(3\pi x), 0, 0]$. The system can thus be reduced to the following controlled

linear system

$$
\begin{bmatrix} \dot{v}_t \\ \dot{v}_r \\ \dot{\sigma}_t \\ \dot{\sigma}_r \end{bmatrix} = \begin{bmatrix} \times & \times & \times & -M_{\rho A}^{-1} D_1^\top \\ \times & \times & -M_{I_\rho}^{-1} D_2^\top & -M_{I_\rho}^{-1} D_0^\top \\ \times & M_{C_b}^{-1} D_2 & \times & \times \\ M_{C_s}^{-1} D_1 & M_{C_s}^{-1} D_0 & \times & \times \end{bmatrix} \begin{bmatrix} v_t \\ v_r \\ \sigma_t \\ \sigma_r \end{bmatrix} + \begin{bmatrix} \times & M_{\rho A}^{-1} B_F \\ M_{I_\rho}^{-1} B_T & \times \\ \times & \times \\ \times & \times \end{bmatrix} \begin{bmatrix} u_\partial^1 \\ u_\partial^2 \end{bmatrix}
$$
(22)

where the mass matrices $M_{\rho A}$, $M_{I_\rho}$, $M_{C_b}$, $M_{C_s}$, matrices $D_0$, $D_1$, $D_2$, vectors $B_F$, $B_T$ are computed through the `fenics` routine and boundary controllers $u_\partial^1$ and $u_\partial^2$ are the control torque and the control force applied at the free end of the beam.

**Stochastic exploration strategy via random walks**  We pre–train the hypersolver model via stochastic exploration of the state-controller space $\mathcal{X} \times \mathcal{U}$. We restrict the boundary control input values $u = [u_\partial^1, u_\partial^2]^\top$ in $[-1, 1] \times [-1, 1]$. As for the state space, naively generating a probability distribution with box boundaries on each of the 160 dimensions of $\mathcal{X}$ would require an inefficient search over this high-dimensional space: in fact, not every combination is physically feasible due to the Timoshenko beam's structure. We solve this problem by propagating batched trajectories with RK4 from the initial boundary condition $z(x, 0) = [\sin(\pi x), \sin(3\pi x), 0, 0]$ with random control actions sampled from a uniform distribution with support in $[-1, 1] \times [-1, 1]$ applied for a time $t_1 \sim U[0.002, 1]$ $s$. We save the states $\{z(x, t_1)^i\}$ and forward propagate from these states again by sampling from the controller and time distributions. We repeat the process $K$ times and obtain a sequence $[\{z(x, t_1)^i\}, \ldots, \{z(x, t_K)^i\}]$ of batched initial conditions characterized by physical feasibility. Finally, we train the hypersolver with stochastic exploration by sampling from the generated distribution $\xi(x, u)$ on local one–step residuals as described in Section 5.4. This initial state generation strategy is repeated every 100 epochs for guaranteeing an extensive exploration of all possible boundary conditions. Figure 13 shows the error propagation over controlled trajectories: the trained HyperEuler achieves the lowest error among baseline fixed–step solvers.

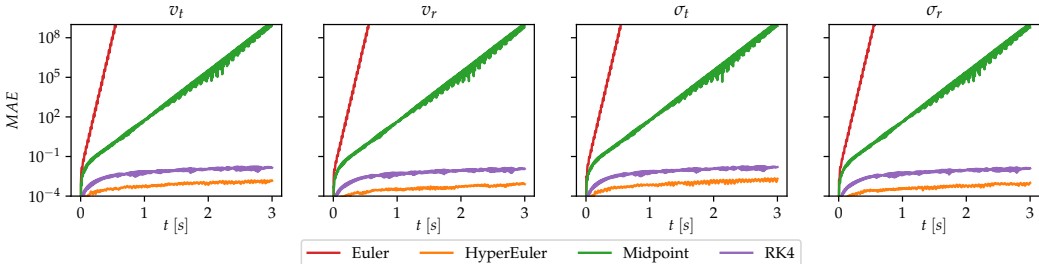

Figure 13: *Mean Absolute Error* (MAE) propagation on velocities $v_t, v_r$ and displacements $\sigma_t, \sigma_r$ for the finite elements of the discretized Timoshenko beam along controlled trajectories. While solutions from Euler and Midpoint quickly diverge due to the system's stiffness, HyperEuler manages not only to contain errors but even outperform the fourth-order RK4 whilst requiring a fraction of the number of vector field evaluations.

**Additional details on the results**  We report additional details regarding the runtime of the experiments on the Timoshenko Beam. As for the training time, training the hypersolver for $10^5$ epochs takes around 80 minutes, where the time for each training epoch slightly varies depending on the length of the sequence of initial condition batches obtained via random walks. As for the averaged runtime per training epoch during the control policy optimization, HyperEuler takes $(2.53 \pm 0.09)$ $s$ per training iteration, Euler $(2.01 \pm 0.04)$ $s$, Midpoint $(4.02 \pm 0.08)$ $s$ and RK4 $(8.24 \pm 0.14)$ $s$. Experiments were run on the CPU of the machine described in Section C.7.

## C.7  HARDWARE AND SOFTWARE

Experiments were carried out on a machine equipped with an AMD RYZEN THREADRIPPER 3960X CPU with 48 threads and two NVIDIA RTX 3090 graphic cards. Software–wise, we used PyTorch (Paszke et al., 2019) for deep learning and the torchdyn (Poli et al., 2020b) and

`torchdiffeq` (Chen et al., 2019) libraries for ODE solvers. We additionally share the code used in this paper and make it publicly available on Github[4].

---

[4]Supporting reproducibility code is at
https : //github.com/DiffEqML/diffeqml − research/tree/master/hypersolvers − control

