# OpenReview forum: "Neural Solvers for Fast and Accurate Numerical Optimal Control"
_ICLR.cc/2022/Conference — ICLR 2022 Poster_

### Official Review · Reviewer_J9x8 · 2021-10-27

**Correctness:** 4
**Technical Novelty And Significance:** 2
**Empirical Novelty And Significance:** 2
**Recommendation:** 5
**Confidence:** 4

**Main Review:**

I thought this paper on the whole was nicely written and easy to understand.

My main issue with the paper is the limited application and its relevance to the ICLR community.
The idea of using NNs to warm start ODE solvers is relevant, but this idea is presented in the paper from Poli et al from Neurips 2020 (althought the paper refence of this work is missing the venue).
However the specific application to optimal control ODEs is somewhat limited and would probably be better placed at a control conference like CDC, ACC or L4DC.
This is reinforced by the experimental settings being quite simple and lwo dimensional.
While ODE-based optimal control is an important setting, the ML community typically use other methods for continuous control.

I also had some minor issues with mathematical clarity. For example the losses in Eq 6 and 6 dont have inputs. In the (unnumbered) equation in 4.2 and Eq. 8 , it is not clear where w enters the objective.

**Summary Of The Paper:**

The paper descibes the application of hypersolves, which warm-start ODE solvers with neural nets that have distilled many precomputed solutions, to optimal control ODEs.
This warm-starting improves downstream accuracy when the computational budget is fixed.
This method is applied for trajectory optimziation for pendulum swing-up and MPC for cartpole and quadcopter systems.

**Summary Of The Review:**

The application of hypersolves to optimal control is somewhat incremental and not of great relevance to the ICLR community. I woudl be willing to change my score if the reviewers and AC disagree.

---

> ### Author Response · Authors · 2021-11-20
> **Response to Reviewer J9x8**
>
> Thank you for your comments and time spent reading our work. Here we would like to address
> the raised concerns:
>
> > My main issue with the paper is the limited application and its relevance to the ICLR community.
> [...] The specific application [of hypersolvers] to optimal control ODEs is somewhat limited and
> would probably be better placed at a control conference like CDC, ACC or L4DC. [...] While ODEbased
> optimal control is an important setting, the ML community typically use other methods for
> continuous control
>
> We would like to express our disagreement with the reviewer’s view about our work’s relevance to
> ICLR community for the following reasons. First of all, since we parameterize the control policy $u_\theta$
> as a neural network, we consider the case of neural optimal control in which a learning-based approach
> is injected in the control task itself similarly to [1] and [2]. Moreover, even papers concerning control methods only without explicitly adding learning components such as [3] have contributed to
> the community by providing a reliable framework for system identification and imitation learning;
> we believe our approach can achieve similar goals. Also, the synergy of learning and control is
> explored in [4] further convincing us that optimal control is closely tied to learning too. Finally, we
> believe our analysis on neural network architectures using [5] and [6] is of relevance to the ICLR
> community. For example, we verified that the hypersolver on the Timoshenko beam with traditional
> activation functions could not obtain a low enough value of error propagation, while it did with the
> $\tt{Snake}$ activation from [5]. Inspired by these results, we believe hypersolvers act as a type of _implicit
> neural representation_ for dynamical systems tied to an ODE solver. Their properties are not only
> limited to the learning of ODE solver residuals in the training region, but extend to generalization to
> unseen states and conditions too. Future works could even extend the generalization properties from
> one system to another, which may bootstrap hypersolver training and downstream control policy
> optimization in various new applications.
>
> > Poli et al from Neurips 2020 [...] reference of this work is missing the venue
>
> We thank the reviewer for pointing out the missing venue on the hypersolver seminal paper. We
> modified the Bibtex citation and added it back along with the link.
>
> > The losses in Eq 6 and 6 dont have inputs
>
> We thank again the reviewer for improving the clarity of our manuscript. We have added input to
> the loss functions in the main text as $\ell(t, x, u)$.
>
> > In the (unnumbered) equation in 4.2 and Eq. 8 , it is not clear where w enters the objective.
>
> We thank the reviewer for pointing out the missing equation numbering - we fixed it in the new
> revision of the paper. Regarding the equation in 4.2 and Equation 8 (that got renumbered to 8 and
> 9 respectively) we did not express the loss  $\ell (t, x, u)$ directly as a function of the hypersolver $g_w$ on
> purpose since $\ell$ can be calculated by either residual fitting, in which case $g_w$ enters the objective
> directly in $\ell$ as per (6) or by trajectory fitting (7) (both in Section 4.1). In the case of trajectory
> fitting, we remind that $x_{k+1}$ can be calculated with (4) for hypersolvers or (12) for multi-stage
> hypersolvers; both these equations contain the optimization parameter $w$.
>
> References
>
> [1] Chen, Y., Shi, Y. and Zhang, B., 2018. Optimal control via neural networks: A convex approach.
> ICLR 2019.
>
> [2] Holl, P., Koltun, V. and Thuerey, N., 2020. Learning to control PDEs with differentiable physics.
> ICLR 2020.
>
> [3] Amos, B., Rodriguez, I.D.J., Sacks, J., Boots, B. and Kolter, J.Z., 2018. Differentiable mpc for
> end-to-end planning and control. NeurIPS 2018.
>
> [4] Jin, W., Wang, Z., Yang, Z. and Mou, S., 2019. Pontryagin differentiable programming: An
> end-to-end learning and control framework. NeurIPS 2020.
>
> [5] Sitzmann, V., Martel, J., Bergman, A., Lindell, D. and Wetzstein, G., 2020. Implicit neural
> representations with periodic activation functions. NeurIPS 2020.
>
> [6] Ziyin, L., Hartwig, T. and Ueda, M., 2020. Neural networks fail to learn periodic functions and
> how to fix it. NeurIPS 2020.

---

> > ### Comment · Reviewer_J9x8 · 2021-11-29
> > **Response**
> >
> > I wish to thank the authors for their additional details.
> >
> > I've gone through the paper again and read the other reviews.
> >
> > Regarding revelence to ICLR, the issue is not the topic of optimal control. Optimal control is relevant, and the papers cited, involving differentiable optimal control algorithms are relevant. The paper looks at improving cheap solvers for gains in computation time, which for me is more applied to the application of these methods and therefore less relevant.
> >
> > Moreover, the paper focuses on optimizing rather small networks, with elaborate search strategies, for really very many epochs (10^5-10^6!) for very simple tasks like pendulum and cartpole swing-up. The line
> >
> > > Keeping their neural network small enough guarantees that evaluating the hypersolvers is cheaper than resorting to more complex numerical routines.
> >
> > suggests that these small networks are crucial for the relevance of this method.
> > For final goal, beating Euler intergration and matching RK4 for the gradient-based OC method, is a bit underwhelming given all the complexity required. When re-reading the paper, I asked myself if the paper was primarily solving a control problem relevant to the ICLR community or demonstrating an application of hypersolvers for control. My personal feeling is that it is the latter, which is why I still believe this work is better suited for a control-oriented venue than ICLR.

---

> > > ### Author Response · Authors · 2021-12-03
> > > **Response to Reviewer J9x8**
> > >
> > > Thank you for your engagement. However, we disagree with the reviewer's personal stance on the relevance of this work to the ICLR community. We elaborate on this point below:
> > >
> > > > When re-reading the paper, I asked myself if the paper was primarily solving a control problem relevant to the ICLR community or demonstrating an application of hypersolvers for control. My personal feeling is that it is the latter, which is why I still believe this work is better suited for a control-oriented venue than ICLR.
> > >
> > > While one of our objectives has indeed been to extend the range of applicability of hypersolvers to controlled systems, the empirical analysis provided extends to aspects that are squarely learning-focused, such as generalization properties of various activations functions in hypersolvers. This work can be framed into the broader class of _implicit neural representations_ - in our case interwoven with an analytical numerical method - which we deem of relevance to the ICLR community, as well as an active research area. We also point out that the latest experiment on PDE control showcases the applicability of our approach even for higher-dimensional systems, in this case a $160$ finite elements discretization of a PDE.
> > >
> > > We remain available to answer further questions, and we hope the reviewer can take into account the above as well as other reviewer perspectives in finalizing their score.

---

### Official Review · Reviewer_RBHY · 2021-11-04

**Correctness:** 4
**Technical Novelty And Significance:** 2
**Empirical Novelty And Significance:** 4
**Recommendation:** 8
**Confidence:** 3

**Main Review:**

Strengths: I thought the paper was a crisp and clear read, and that all technical components (both novel and tutorial) and all experiments were both sound and very well-explained.

Weaknesses: The systems that are tested on are relatively small scale/low-dimensional. Given that hypersolver frameworks are presumably most necessary in larger scale systems that are indeed expensive to solve, this is in my view a large limitation in terms of understanding the value proposition of the present work. It would greatly strengthen the present work if demonstrations on such large-scale/high-dimensional systems could be shown.

Minor points (not affecting my score):
* $\psi_\epsilon$ in Equation (3) could be more explicitly defined, as it is of course a critical variable for the rest of the paper.
* The differentiation of the present work from the prior hypersolvers work could be made more clear, and in general the writing could better signpost what aspects of the present work are "new" contributions. If I understand correctly, the major distinction from the prior work is that the prior work considered uncontrollable systems, whereas the present work considers controllable systems, which yields some differences both in terms of how the hypersolver is trained and in how it is used. However, this should be spelled out more explicitly.

**Summary Of The Paper:**

This paper extends the hypersolvers framework (Poli et al., 2020a) to the setting of optimal control. The hypersolvers framework combines a cheap numerical dynamical system solver with a neural network trained to approximate the truncation error of this solver, which together yield a cheap and relatively accurate solver. This paper shows how to utilize the hypersolvers framework within a controlled dynamical system by exploring different choices of loss functions, training methods, and network architectures for hypersolver pre-training. The authors further provide an augmentation to this framework, called multi-stage hypersolvers, to further account for misspecifications of the dynamics. Via experiments, the authors then demonstrate the performance of optimal control strategies (direct optimal control and MPC) utilizing hypersolvers in different settings (notably the pendulum, cart-pole, and quadcopter settings). They show e.g. that even in cases where their method receives misspecified dynamics, they are able to achieve control performance that is comparable to that obtained in cases where the underlying solver had knowledge of the correct dynamics.

**Summary Of The Review:**

I thought this paper was a sound, enjoyable, and clear extension of the previous work on hypersolvers to the setting of controlled dynamical systems. However, providing experiments on larger-dimensional systems would in my view significantly strengthen the paper by more fully demonstrating the potential benefits of the proposed method. That said, I am marginally inclined to accept the paper nonetheless rather than rejecting it solely on that basis.

---

> ### Author Response · Authors · 2021-11-20
> **Response to Reviewer RBHY**
>
> We thank the reviewer for the invaluable feedback and for acknowledging the strengths of our work.
> We address concerns and clarify some points in the sections below.
>
> > The systems that are tested on are relatively small scale/low-dimensional. Given that hypersolver
> frameworks are presumably most necessary in larger scale systems that are indeed expensive to
> solve, this is in my view a large limitation in terms of understanding the value proposition of the
> present work.
>
> We agree with the reviewer that the experiments we have previously considered are relatively small
> scale. However, we believe that control policy optimization on these non–linear dynamical systems is
> still a non-trivial task. Indeed, experiments in the literature for learning and control have considered
> similar settings as ours as their main benchmarks. For example, the inverted pendulum and cart-pole
> are the only dynamical systems considered in the experiments of [1] while the cart-pole and
> quadcopter environments in [2] are fundamental benchmarks for comparisons to provide empirical
> evidence of the framework presented in their work.
>
> > It would greatly strengthen the present work if demonstrations on such large-scale/high-dimensional
> systems could be shown.
>
> Based on the reviewer's input, we added a new experimental setting for demonstrating hypersolver scaling on a high–dimensional
> system, namely the discretized model of the Timoshenko beam of [3]. The system, described in its
> continuous formulation via a partial differential equation (PDE), is first discretized into 40 finite elements
> via a PDE runtime. The obtained system consists of a state of 160 dimensions (an increase of
> more than $10 \times$ in the number of dimensions compared to the previous highest–dimensional system, the quadcopter). We trained a hypersolver to speed up the optimization of a boundary control policy
> that aims at stabilizing the beam in the straight position. Results in terms of error propagation,
> learned control policy and runtime costs compared to other baseline solvers show promising results
> and demonstrate how hypersolvers for optimal control can scale up and obtain accurate and efficient
> solutions.
>
> > $\psi_\epsilon$ in Equation (3) could be more explicitly defined, as it is of course a critical variable for the rest
> of the paper.
>
> In the paper, we focus on the analysis of the HyperEuler scheme, in which $\psi_\epsilon \left(t_k, x_{k},u_k \right) = f \left(t_k, x_{k},u_k \right)$ , i.e. the first-order derivative of the system. The HyperEuler and multi-stage HyperEuler
> scheme is explicitly defined in Sections A.1 and A.2 of the Appendix. We have added to
> the main text links to these Sections at the first occurrences of the HyperEuler schemes so that the
> variable  $\psi_\epsilon$ can be clearer to the audience.
>
> > The differentiation of the present work from the prior hypersolvers work could be made more clear,
> and in general the writing could better signpost what aspects of the present work are ”new” contributions.
> [...] This should be spelled out more explicitly.
>
> We have added new references and emphasized other main differences from the previous works in
> Related Works (Section 6).
>
> > If I understand correctly, the major distinction from the prior work is that the prior work considered
> uncontrollable systems, whereas the present work considers controllable systems, which yields some
> differences both in terms of how the hypersolver is trained and in how it is used.
>
> The reviewer is correct: the main contribution of our work is the development of a two–stage framework
> which can be summarized in:
> 1. Hypersolver architecture design and pre–training methods via different explorations strategies
> of the state–controller space,
> 2. Subsequent use of the hypersolver for solving numerical
> optimal control problems accurately and efficiently.
>
> Finally, given that we addressed the reviewer’s primary concern raised in the review, we would
> kindly ask to adjust the review score while taking the rebuttal into account. If any additional concerns
> or questions should arise, we will be more than glad to address them.
>
> References
>
> [1] Amos, B., Rodriguez, I.D.J., Sacks, J., Boots, B. and Kolter, J.Z., 2018. Differentiable MPC for
> end-to-end planning and control. NeurIPS 2018.
>
> [2] Jin, W., Wang, Z., Yang, Z. and Mou, S., 2019. Pontryagin differentiable programming: An
> end-to-end learning and control framework. NeurIPS 2020.
>
> [3] Macchelli, A. and Melchiorri, C., 2004. Modeling and control of the Timoshenko beam. The
> distributed port Hamiltonian approach. SIAM Journal on Control and Optimization, 43(2), pp.743-
> 767.

---

> > ### Comment · Reviewer_RBHY · 2021-11-20
> > **Good response**
> >
> > Thanks to the authors for adding the additional (higher-dimensional) experiment. As the authors indeed addressed my chief concern, I have now raised my score.

---

### Official Review · Reviewer_jwFx · 2021-11-04

**Correctness:** 4
**Technical Novelty And Significance:** 3
**Empirical Novelty And Significance:** 2
**Recommendation:** 6
**Confidence:** 4

**Main Review:**

The paper proposes to apply the idea of hypersolvers to numerical optimal control. Through learning the truncation residual dynamics, the model is able to approximate the actual dynamics with a combination of low-order ODE solver with learned truncation residual dynamics. Further, if the system model is not perfectly accurate, a multi-stage hypersolver can be used to separately learn the system residual dynamics and truncation residual dynamics. In experiments, the method is shown to perform on par with the higher-order ODE solvers. Also, experiments show that certain activation functions (SIREN and Snake) perform better than the standard ones (Tanh and ReLU).

Here are some of my detailed comments:
1. In (3), \epsilon appeared before it is introduced in (4), hence making it undefined and unclear.
2. In section 3.2, maybe more explanation can be given on how (2) is solved?
3. In the pendulum experiment, the hypersolver approach achieves similar performance in terms of final positions when compared with Midpoint and RK4, but using similar number or 50% less of FLOPS. For the other two experiments, I don’t seem to find how does the method compares in terms of FLOPS or runtime. As in the pendulum case, the proposed hypersolver approach does not outperform a lot in terms of runtime or FLOPS.
4. In the pre-training of the hypersolvers, it seems to require a very large number of epochs, maybe more information can be provided on how many trajectories are used for training and how long the training time is.
5.  In section 5.2, how does the tolerance of the ODE solver affect the experiment results, maybe some ablation studies and explanations can be included to help understand.
6. For the multi-stage hypersolvers, there are some literature on learning the system residual dynamics for control, such as [1][2][3]. Also some ablation studies can be done by comparing the multi-stage hypersolver approach with just learning system residual dyanamics.

[1] Saveriano, Matteo, Yuchao Yin, Pietro Falco, and Dongheui Lee. "Data-efficient control policy search using residual dynamics learning." In 2017 IEEE/RSJ International Conference on Intelligent Robots and Systems (IROS), pp. 4709-4715. IEEE, 2017

[2] J. F. Fisac, A. K. Akametalu, M. N. Zeilinger, S. Kaynama, J. Gillula, and C. J. Tomlin. A general safety framework forlearning-based control in uncertain robotic systems.IEEE Transactions on Automatic Control, 64(7):2737–2752,2018.

[3] A. Taylor, A. Singletary, Y. Yue, and A. Ames. Learning for safety-critical control with control barrier functions. InLearning for Dynamics and Control, pages 708–717. PMLR, 2020.


**Summary Of The Paper:**

In this paper, the authors apply the idea of hypersolvers to numerical optimal control. The idea is to approximate the dynamics with a low-order ODE solver and learn the truncation residual dynamics using a neural net. They also propose a multi-stage hypersolvers method to both learn the system residual dynamics (due to erroneous system model) and truncation residual dynamics (due to ODE solver). The performance of the proposed approach is shown in simulation experiments with a pendulum, a cart-pole, and a quadcopter with comparisons to ODE solvers of different orders.

**Summary Of The Review:**

Overall, the paper proposes to use hypersolvers to speed up ODE solving for numerical control. The idea is promising but it needs more experiment results comparisons, especially in terms of runtime, to justify the efficacy of using hypersolvers for optimal control. Also, the paper could consider expanding the explanation on how the optimal control problem (2) is solved in their experiments.

---

> ### Author Response · Authors · 2021-11-20
> **Response to Reviewer jwFx (Part 1/2)**
>
> We thank the reviewer for the thoughtful and insightful review. Below, we do our best to address the
> reviewer’s questions.
>
> > In (3), $\epsilon$ appeared before it is introduced in (4), hence making it undefined and unclear.
>
> Thanks for pointing this out. The step size is an essential part of numerical solvers and clarifications
> on this aspect are important for understanding our method. As a result, in the revised version of the
> manuscript, we specified the meaning of step size $\epsilon$ in (3) right after the equation to make it clearer.
>
> > In section 3.2, maybe more explanation can be given on how (2) is solved?
>
> We included a new Section B.3 in Appendix which expands the explanation of Section 3.2 about the
> solution of (2). Namely, we consider the case of _neural optimal control_ in which we parametrize
> the controller  $u_\theta: t, x \mapsto u_\theta(t, x)$ as a multi–layer perceptron. We then minimize the control cost
> function of Section B.1 with stochastic gradient descent by using the ${\tt Adam}$ optimizer.
>
> > In the pendulum experiment, the hypersolver approach achieves similar performance in terms of
> final positions when compared with Midpoint and RK4, but using similar number or 50% less of
> FLOPS. For the other two experiments, I don’t seem to find how does the method compares in
> terms of FLOPS or runtime. As in the pendulum case, the proposed hypersolver approach does not
> outperform a lot in terms of runtime or FLOPS.
>
> The reviewer is correct regarding the FLOPs count: HyperEuler achieves similar performance to e.g.
> the Midpoint, using a little less than 50% of the FLOPs in the pendulum experiment. However, this result is an important indicator
> for the performance of the hypersolver which still considerably improves on the Euler baseline in
> terms of solution accuracy. This in turn provides better solutions for the downstream numerical
> optimal control problem optimization. In the cart-pole system, the main purpose of the multi-stage
> hypersolver was to not only model higher-order solver residuals but also to correct dynamics. For
> the quadcopter experiment, local solution accuracy is higher than Midpoint and the control policy
> optimized with HyperEuler manages to even outperform the fourth-order Runge Kutta.
>
> We also included a new experiment to investigate hypersolver with high–dimensional controlled
> systems in the revised version of our manuscript, namely the finite element discretization of the
> Timoshenko beam’s PDE resulting in a 160–dimensional system. We trained the hypersolver on it
> and optimized a boundary controller with the task to keep the beam straight. Experimental results
> showed how HyperEuler could obtain a similar performance with RK4 while Euler and Midpoint
> failed the control task. Based on the reviewer’s input, we also provided FLOPs counts and runtime
> for the experiment, which showed more than $3 \times$ speedups in the control policy training time. This
> experiment showcased how hypersolvers scale up to larger-scale systems the problem and are even
> more impactful in high–dimensional regimes.

---

> > ### Author Response · Authors · 2021-11-20
> > **Response to Reviewer jwFx (Part 2/2)**
> >
> > > In the pre-training of the hypersolvers, it seems to require a very large number of epochs, maybe
> > more information can be provided on how many trajectories are used for training and how long the
> > training time is.
> >
> > The pre-training of hypersolvers requires a thorough exploration of the state-controller space $\mathcal{X} \times \mathcal{U}$.
> > We note that the distribution of initial conditions $\xi(x,u)$ can produce an infinite number of different
> > initial conditions due to the continuous nature of the control problems we consider, unlike most
> > datasets which contain a finite number of elements. Thus, even though the number of epochs is large
> > in absolute value, the time required for each epoch is relatively low. To put that into perspective, for
> > the 160–dimensional Timoshenko beam experiment, training is performed at a speed of more than 20 epochs per second. For the other experiments, hundreds of epochs are processed every second.
> > Moreover, our approach has access to the dataset generators themselves, i.e., the system dynamics.
> > This allows us not only to efficiently generate training data _online_ during the training process itself,
> > but also to save memory by not needing to store datasets.
> >
> > > In section 5.2, how does the tolerance of the ODE solver affect the experiment results, maybe some
> > ablation studies and explanations can be included to help understand.
> >
> > This is an interesting point raised by the reviewer. In principle, the absolute and relative tolerances
> > of the ODE solver set the maximum absolute error allowed in a solution and the maximum error
> > relative to the solution value respectively. Lower tolerances usually trade better accuracy for a
> > longer computation time. We investigated this property and tried training hypersolvers with different
> > combinations of absolute and relative tolerances in the range $[10^{-4}, 10^{-7}]$ and tested them on local
> > residual values. However, the tests showed no correlation between local errors and absolute/relative
> > tolerances. We hypothesize the reason behind this result is that the differential equations describing
> > the controlled systems we used are not too stiff and as a result different values of tolerances do not
> > change solutions in any noticeable way. On the other hand, setting tolerance values lower than $10^{-7}$
> > resulted in a considerably larger training time (even more than 100 times slower), which made it
> > time–consuming and impractical to train hypersolvers. Therefore, we decided to resort for tolerance
> > values in the range considered $[10^{-4}, 10^{-7}]$ for the Tsitouras45 adaptive–step solver.
> >
> > > For the multi-stage hypersolvers, there are some literature on learning the system residual dynamics
> > for control, such as [2][3][4]. Also some ablation studies can be done by comparing the multi-stage
> > hypersolver approach with just learning system residual dynamics.
> >
> > We thank the reviewer for sharing insights and related work. We included the papers and their references
> > with their relation to multi-stage hypersolvers in the Related Works (Section 6). Moreover,
> > we included a new paragraph in Section C.4 of the Appendix in which learning residual dynamics
> > only is compared to other baseline solvers on inaccurate and accurate dynamics and the multi-stage
> > hypersolver approach.

---

### Author Response · Authors · 2021-11-20
**Response to All Reviewers - Manuscript Revision Summary**

We want to thank all the reviewers for taking the time and effort to read our paper thoroughly
and for providing insightful and productive comments, both on the theoretical and experimental
aspects of the paper. We summarize the revisions we have made in response to the reviews below.

### New experiment

We added a new experiment to demonstrate how hypersolvers
scale in high–dimensional regimes, namely the boundary control of the Timoshenko beam
from [1]. The resulting system is a finite element discretization with 40
elements of the Timoshenko PDE for a total of 160 state dimensions. We pre-train a hypersolver,
optimize a boundary control policy, and compare the results with other baseline solvers. We
further provide additional details regarding the runtime for hypersolver training and control policy
optimization of this experiment (thanks to Reviewers RBHY and J9x8 for encouraging us in tackling
high–dimensional control settings and to Reviewer jwFx to urge us to include runtime comparisons).

### Main text additions

We introduced clarifications and more details as below:

1. Included new literature references [2][3][4] to Section 6 and expanded the explanation
around learning residual dynamics and our contributions (thanks to Reviewers jwFx and
J9x8)
2. Added new Section B.3 in Appendix further expanding the explanation of Section 3.2 about
how we solve problem (2) with neural optimal control (thanks to Reviewer jwFx)
3. Added new paragraph under Section C.4 in the Appendix. We compare the multi-stage
hypersolver with other baseline solvers on inaccurate and accurate dynamics and learning
residual dynamics only (thanks to Reviewer jwFx)

### Clarity improvements

We made changes to some mathematical notation:

1. Step size $\epsilon$ in (3) is now specified right after the equation (thanks to Reviewer jwFx)
2. Missing numbering was added for the Equation in Section 4.2, which is now Equation 8
(thanks to Reviewer J9x8)
3. Added inputs for Losses in Equations 6 and 7 (thanks to Reviewer J9x8)

### Miscellaneous changes
We made some minor changes to accommodate the new material:

1. Table 1 now includes the new experiment as well. Labels were slightly modified to fit the
parameters in a single table
2. Old Figure 6 was split into two parts since it provides different visualizations of the same
experiment (phase space and state propagation in time). The visualization of state propagation
in time was moved to Section C.3 in the Appendix and is now Figure 10
3. Old Figure 7 (Top) was moved to Section C.4 in the Appendix and is now Figure 12. Also,
it now shows error propagation of the Euler solver with learning residual dynamics
4. Added missing NeurIPS venue and link for the seminal hypersolver reference (thanks to
Reviewer J9x8)

### New References

[1] Macchelli, A. and Melchiorri, C., 2004. Modeling and control of the Timoshenko beam. The distributed port Hamiltonian approach. SIAM Journal on Control and Optimization, 43(2), pp.743-767.

[2] Saveriano, Matteo, Yuchao Yin, Pietro Falco, and Dongheui Lee. ”Data efficient
control policy search using residual dynamics learning.” In 2017 IEEE/RSJ International
Conference on Intelligent Robots and Systems (IROS), pp. 4709-4715. IEEE, 2017

[3] J. F. Fisac, A. K. Akametalu, M. N. Zeilinger, S. Kaynama, J. Gillula, and C. J. Tomlin. A
general safety framework for learning-based control in uncertain robotic systems.IEEE Transactions
on Automatic Control, 64(7):2737–2752,2018

[4] A. Taylor, A. Singletary, Y. Yue, and A. Ames. Learning for safety-critical control with control
barrier functions. In Learning for Dynamics and Control, pages 708–717. PMLR, 2020.

---

### Decision · Program_Chairs · 2022-01-20

**Decision:**

Accept (Poster)

**Comment:**

The authors propose a novel hypersolver framework for solving numerical optimal control problems, learning a low order ODE and a neural network based residual dynamics. They compare their framework with traditional optimal control solvers on a number of control tasks and demonstrate superior performance.

The reviewers are in consensus that the paper makes significant contributions that are validated by the experimental results. The only concern was that the experiments are largely on low dimensional systems, but the reviewers agreed that the results are still worthy of acceptance.